# Interaction of Destruxin A with Three Silkworm Proteins: BmCRT, BmDPP3, and BmPDIA5

**DOI:** 10.3390/molecules27227713

**Published:** 2022-11-09

**Authors:** Xuyu Yin, Haitao Peng, Qunfang Weng, Qiongbo Hu, Jingjing Wang

**Affiliations:** 1Key Laboratory of Bio-Pesticide Innovation and Application of Guangdong Province, College of Plant, Protection, South China Agricultural University, Guangzhou 510642, China; 2College of Horticulture, South China Agricultural University, Guangzhou 510642, China

**Keywords:** Destruxin A, BmCRT, BmDPP3, BmPDIA5, binding protein

## Abstract

Destruxin A (DA), a hexa-cyclodepsipeptidic mycotoxin produced by the entomopathogenic fungus *Metarhizium anisopliae*, has insecticidal activity, but its molecular mechanism of action is still not clear. Three proteins with modification-related functions, calreticulin (BmCRT), dipeptidyl peptidase Ⅲ (BmDPP3), and protein disulfide isomerase A5 (BmPDIA5), were selected to verify the interactions with DA in this study. The kinetic data of the interactions were measured by surface plasmon resonance (SPR) and bio-layer interferometry (BLI) in vitro. The K_D_ values of DA with BmCRT, BmDPP3, and BmPDIA5 ranged from 10^−4^ to 10^−5^ mol/L, which suggested that the three proteins all had fairly strong interactions with DA. Then, it was found that DA in a dose-dependent manner affected the interactions of the three proteins with their partners in insect two-hybrid tests in SF-9 cells. Furthermore, the results of enzyme activities by ELISA indicated that DA could inhibit the activity of BmDPP3 but had no significant effect on BmPDIA5. In addition, DA induced the upregulation of BmDPP3 and the downregulation of BmCRT. The results prove that BmCRT, BmDPP3, and BmPDIA5 are all binding proteins of DA. This study might provide new insights to elucidate the molecular mechanism of DA.

## 1. Introduction

Destruxins are one of the most potent toxins, which can be synthesized by different species of entomopathogenic fungi *Metarhizium anisopliae*, *Lecanicillium longisporum*, and *Aschersonia* sp. [1], and five natural analogues (labeled A–E) have been isolated [2]. Destruxin A (DA, Figure 1), which has 39 analogues, secreted by the entomopathogenic fungus *Metarhizium anisopliae*, is a cyclodepsipeptidic mycotoxin with vigorous insecticidal activity and anti-immunity effects on more than 20 kinds of host insects [3]. As a considerable insecticidal mycotoxin, DA was reported to affect muscle contraction [4], damage insect tissues such as the neuromuscular system, midgut, and Malpighian tubules [1,5], bring on equilibrium chaos of intra- and extra-cellular hydrogen and calcium ion in *Bombyx mori* [6,7], and regulate immune related gene expression [8]. Moreover, in the past decade, several studies indicated that DA changes the morphology of hemocytes and subsequently affects the function of phagocytosis and encapsulation [6,7]. DA is considered a new potential pesticide, attracting researcher interest. The interactions of DA with proteins have been little studied. The most recent study of them shows that the interactions of DA with arginine tRNA synthetase, lamin-C proteins, and aminoacyl tRNA synthases in *Bombyx mori* [9,10]. However, the research on the insecticidal mechanism of DA has not been in-depth and systematic enough, and its mechanism of action has not been clarified [9,10]. To gain more insight into the mode of action of DA, in this study, three proteins, calreticulin (BmCRT), dipeptidyl peptidase Ⅲ (BmDPP3), protein disulfide isomerase A5 (BmPDIA5), were selected from nearly 100 potential DA-binding proteins based on our previous research to study the interactions between DA further.

As a molecular chaperone of the endoplasmic reticulum, BmCRT binds calcium ions, maintains the calcium balance in cells, regulates the expression of genes, and directs the proper folding of proteins and glycoproteins [11]. BmCRT also participates in immune regulatory response, and plays a role in wound healing, phagocytosis, and encapsulation [12,13,14]. In addition, it is involved in cell adhesion, lectin-like chaperoning, and so on [15,16]. BmDPP3 is a zinc-dependent metalloproteinase of the M49 family, with serine as its active center [17]. It has been associated with protein turnover, grading oligopeptides, and the metabolism of proteins [18]. Furthermore, it has also been implicated in pain modulation, blood pressure regulation, inflammatory response, and the cytoprotective effect against oxidative stress [19]. BmPDIA5 serves as a molecular chaperone of the endoplasmic reticulum, and catalyzes the formation, fracture, reduction, rearrangement, and isomerization of disulfide bonds [20]. It can catalyze client folding and domain/subunit assembly through its distinct functions and structures and prevent non-productive folding aggregation [21]. Moreover, BmPDIA5 has been reported to play essential roles in embryonic development or function of the brain and central nervous systems [22]. In this study, we validated the interaction between DA with these proteins in vivo and in vitro by multiple experimental methods. The results may provide new insights into the mechanism of action of DA.

## 2. Results

### 2.1. Binding Affinity of DA with Three Proteins Determined by Surface Plasmon Resonance (SPR) and Bio-Layer Interferometry (BLI) In Vitro

After gene cloning and vector construction, three silkworm proteins were expressed and purified in vitro, and then the recombinant proteins were detected in SDS-PAGE (Figure 2A). On SPR, the binding affinity of DA with these proteins was determined by SPR, and the affinity of DA and BmCRT, BmDPP3, and BmPDIA5 was 1.98 × 10^−4^ mol/L, 1.81 × 10^−4^ mol/L, 9.26 × 10^−5^ mol/L, respectively (Figure 2B–D). On BLI, the affinity of DA and BmDPP3, BmPDIA5 was 5.87 × 10^−5^ mol/L, 4.47 × 10^−5^ mol/L (Figure 2E,F). BmCRT cannot be measured by BLI because it cannot be connected to the chip. The K_D_ of three proteins to DA ranged from 10^−4^ to 10^−5^ mol/L, indicating interactions between DA and three proteins with moderate intensity.

### 2.2. The Interactions between DA and Three Proteins Determined by Insect Two-Hybrid (I2H) In Vivo

The effects of DA on three proteins in cells were studied by I2H. The interacting proteins of the above three proteins were found through the String protein database and related literatures. BmCRT is the interacting protein of BmPDIA5 [23], and the interacting protein of BmDPP3 is BmKEAP1 [24]. We needed to verify first whether there were interactions between them before studying the effects of DA on protein interactions. The luminescence values of *Spodoptera frugiperda* 9 (Sf9) cells transfected with I2H target vectors containing two groups of proteins were significantly different from the cells not transfected (*p* < 0.05) (Figure 3), which means the genes were expressed. Moreover, there were interactions between BmDPP3 and BmKEAP1, BmCRT, and BmPDIA5.

The interactions of the above two groups of proteins were affected under the DA treatment, showing a DA concentration-dependent effect (Figure 4). The relative luminescence value (DA treatment group/CK group) was analyzed. The interactions of two groups of proteins were significantly reduced when the DA concentration was more than or equal to 0.2 mg/L, which suggested noticeable inhibitory effects. The relative luminescence values were below 0.5 when the DA concentration was more significant than or equal to 2 mg/L, and there was a more significant reduction of BmCRT-BmPDIA5, the relative luminescence value of which was 0.3, meaning a more obvious inhibitory effect. When the concentration of DA was less than or equal to 0.02 mg/L, there was no effect on the interactions of both groups of proteins.

### 2.3. Effects of DA on the Enzymatic Activities of BmDPP3 and BmPDIA5

DA affects the activity of BmDPP3 (Figure 5), and there was no difference with CK (*p* < 0.05) when the DA concentration was less than or equal to 0.1 mg/L, but when the concentration of DA was more significant than or equal to 1 mg/L, the enzymatic activity of BmDPP3 was significantly reduced (*p* < 0.05).

DA has no effect on the activity of BmPDIA5 (Figure 6). There was no significant difference with CK (*p* < 0.05) when the DA concentration was less than or equal to 100 mg/L, indicating that DA had no inhibitory effect on BmPDIA5 in this concentration range. However, BmCRT has no enzymatic activity.

### 2.4. Effects of DA on the Gene Expression of Three Proteins

Bm12 cells were treated with DA for 8 h and detected the expression of three genes. The results of qPCR showed that the gene expression of BmDPP3 was upregulated by approximately 109% (relative expression of 2.09 times), the gene expression of BmCRT was downregulated by 34% (relative expression of 0.66 times), and there was no significant change of BmPDIA5 (Figure 7). The expression of BmCRT, BmDPP3, and BmPDIA5 were not significantly affected by DA at lower concentrations.

## 3. Discussion

BmCRT and BmPDIA5 are both molecular chaperones in the endoplasmic reticulum. BmCRT can promote the interaction between proteins, repair the misfolding of proteins and glycoproteins, and direct proper folding of them [25], such as calreticulin functions in recognition of misfolded MHC class I heavy chains in the endoplasmic reticulum [26]. BmPDIA5 can catalyze the formation, fracture, and rearrangement of disulfide bonds, and involve the folding of proteins and glycoproteins [22]. For example, BmPDIA5 contributes to disulfide bond rearrangement in activating transcription factor 6 alpha, and it is a specialized member that participates in the folding of α1-antitrypsin and N-linked glycoproteins [20,27]. BmDPP3, a zinc-dependent metalloproteinase, plays a role in the turnover and metabolism of proteins [28]. We speculated that DA could affect the synthesis and degradation of protein and interfere with the normal life activities in silkworms. In addition, DA may take effect by disrupting the homeostasis in silkworms [11]. BmCRT can maintain the calcium balance in cells and regulate gene expression. BmDPP3 protects oxygen-glucose deprivation/reoxygenation-injured neurons by suppressing apoptosis, oxidative stress, and inflammation via modulation of Keap1/Nrf2 signaling [19]. BmPDIA5 can alleviate and repair the damage to cells and tissues caused by faulty and incomplete folding of proteins [29]. The interactions between protein and DA may inactivate the protein, destroy the homeostasis and balance in silkworms, interfere with the physiological function, and affect their normal life activities.

The interactions between BMDPP3 and BMKEAP1, and BMCRT and BMPDIA5 were significantly inhibited by DA dosage-dependent manner, indicating that there were interactions between DA and three proteins. The binding of DA to the proteins changes the conformation of the proteins, or DA directly competes for the binding sites with the binding proteins, which leads to the reduction of the interaction intensity between proteins. The effects of DA on the protein activities of BmDPP3 and BmPDIA5 were different, and the experimental data showed that DA could affect the activity of BmDPP3, but hardly affect the activity of BmPDIA5. We speculated that the binding site of DA to BmDPP3 in vitro was its active site, and the binding of DA to BmDPP3 decreased the activity of BmDPP3, while BmPDIA5 did not. The binding of BmDPP3 to DA changes the conformation and function of BmDPP3, so the exploration of the changes of BmDPP3 structure sites before and after the binding of BmDPP3 to DA can provide directions for the search of the active site of BmDPP3.

It has been found that insects treated with DA are more susceptible to disease, and DA treatment can also improve the insecticidal activity of chemical agents. At the same time, the three proteins are all related to immune function. BmCRT can function as lectin, and can promote wound healing, regulate phagocytosis, encapsulation, and so on. In addition, BmCRT plays an important role in cell adhesion, regulating immune response, pathological changes, and anti-virus [11]. BmDPP3 is closely related to the inhibition of inflammatory response and the cytoprotective effect against oxidative stress [19]. BmPDIA5 is also indirectly associated with immune responses. It can be speculated that DA may have an effect on the three proteins or the interaction between BmCRT, and BmPDIA5, affecting some immune-related proteins, thereby reducing the immune function of insects.

In conclusion, we investigated the interaction of DA with three silkworm proteins by SPR, BLI, I2H, enzyme activity analysis, qPCR. As a result, BmCRT, BmDPP3, and BmPDIA5 were all verified to be DA-binding proteins, and it is of great significance for the study of the molecular mechanism of DA.

## 4. Materials and Methods

### 4.1. Cell Culture and DA

The Sf9 cells from our laboratory were cultured in SFX culture medium (Hyclone^TM^, Pittsburgh, MA, USA) with 5% fetal bovine serum (Gibco^TM^, Waltham, MA, USA). Bm12 cells were donated by Professor Cao Yang, College of Animal Science at South China Agricultural University, and cultured in TNM-FH culture medium (Hyclone) with 10% fetal bovine serum (Gibco). Cells were cultured at 27 °C and maintained for 2–4 days. DA was isolated and purified from the *Metarhizium anisopliae* var. anisopliae strain MaQ10 in our laboratory. The DA stock solution of 10,000 µg/mL was prepared from 1 mg DA and 100 µL dimethyl sulfoxide (DMSO, Sigma-Aldrich, Darmstadt, Germany).

### 4.2. The Expression, Isolation and Purification of Proteins

The BL21 (DE3) transformant containing recombinant plasmid was selected and cultured in LB (Kanamycin 50 mg/L, Ampicillin 100 mg/L), then expanded the culture in small quantities and induced the expression by IPTG (final concentration 1 mmol/L), and the protein expression was detected by SDS-PAGE (DYY-6C Electrophoresis Apparatus, Beijing Liuyi Biotechnology Co., Ltd., Beijing, China). After the successful expression, the culture was expanded in large quantities and the expression induced, and the cells were collected by centrifugation (TGL16 High-Speed Refrigerated Centrifuge, Changsha Yingtai Instrument Co., Ltd., Changsha, China). The supernatant and precipitate were separated and collected after the cell disruption by ultrasound. After another detection of protein expression, the inclusion body protein was prepared for purification, renaturation, and SDS-PAGE.

### 4.3. Surface Plasmon Resonance (SPR)

The running buffer of the instrument was configured first. Since DA is insoluble in water, DMSO was needed, and the running buffer was configured as 1 × PBSP (5% DMSO). Then, the different concentrations of DA (250 µmol/L, 125 µmol/L, 62.5 µmol/L, 31.25 µmol/L, 15.625 µmol/L) were configured by the running buffer. The correction solution with the DMSO content of 4.5–5.8% was required to eliminate the interference of DMSO on the experiment of interaction, and the detailed steps are provided in the Biacore 8K (GE, Fairfield, CT, USA) user manual. The experiment can be started after all the solutions have been prepared, and the sample can be injected and run according to the prompts of the instrument. After the operation, the kinetic constants of DA and proteins were calculated by the multi-cycle analysis method of the instrument, and the experiment was repeated at least twice.

### 4.4. Bio-Layer Interferometry (BLI)

BLI analysis was performed on a ForteBio OctetQK System (K2, Pall Fortebio Corp, Menlo Park, CA, USA) and referred to our previous literature [30,31]. Serial, not gradient, dilutions of DA (12.5 µmol/L, 25 µmol/L, 50 µmol/L, 100 µmol/L, 200 µmol/L, 400 µmol/L) were used for treatment. PBST buffer (0.05% Tween20, 5% DMSO) was used for the reference and dilution buffers. Two SSA biosensors were coupled with protein, cured for one h, and the signal height was 1.5 nm. One biosensor is used to balance the buffer solution, and the other combines the dissociation signals of different concentrations of DA to measure the binding force with different concentrations of DA. The experiment was conducted and completed at 1000 rpm, 30 °C, 220 µL per well. Finally, the raw data were processed with Data Analysis Software (9.0, Pall Fortebio Corp, Menlo Park, CA, USA).

### 4.5. Effects of DA on Enzyme Activities of BmDPP3 and BmPDIA5

The enzyme activities of BmDPP3 and BmPDIA5 were determined by the corresponding enzyme activity detection kits (Shanghai Huzhen Industry Co., Ltd., Shanghai, China). The OD values of different samples were determined after the processes of reaction, incubation, color development and termination, etc., which were compared with the measured OD values of the standards to calculate the enzyme activities. After this, enzyme activities were analyzed by one-way ANOVA through the IBM SPSS statistics 24 (IBM, Armonk, NY, USA), and the significance test of differences was performed by Duncan’s new multiple-range method.

### 4.6. Insect Two-Hybrid (I2H)

The interacting proteins of the above three proteins were found through the Protein Interaction Network Database: String. There is an interaction between BmCRT and BmPDIA5, and the interacting protein of BmDPP3 is BmKEAP1. Primers were designed based on the CDS sequences of the four proteins, and PCR (Bio-Rad, Shanghai, China) amplification and purification were performed, then they were ligated into the Gateway clone vector. The entry vectors of the proteins to be tested and their interacting proteins were reacted respectively with the pIE2-AD and pIE2-DBD in LR enzyme to construct the insect two-hybrid expression vectors for the cell transfection. The constructed vectors were mixed with a transfection agent and added to SF9 cells after the incubation. After the transfection, the cells were treated with different concentrations of DA (final concentrations 0.002 mg/L, 0.02 mg/L, 0.2 mg/L, 2 mg/L, 20 mg/L) and incubated. After the treatment with luciferase reaction reagent (Promega, Madison, WI, USA) in white 96-well plates (Corning Incorporated, Corning, NY, USA), the luminescence values were detected by Synergy^TM^ H1 (BioTek, Winooski, VT, USA), and the data were analyzed by SPSS.

### 4.7. Effects of DA on the Gene Expression of Three Proteins

Bm12 cells in the logarithmic phase were inoculated into 12-well plates and cultured for 12 h. Different concentrations of DA (final concentrations 0.02 mg/L, 0.2 mg/L, 2 mg/L) were added to the experimental groups, and control was set up. Total RNA was extracted and tested for its purity and concentration after 8 h, and then RNA was used as the template to synthesize the cDNA for the polymerase chain reaction. The silkworm GAPDH (glyceraldehyde-3-phosphate dehydrogenase) gene was the reference gene. The qPCR reaction system consisted of 1 µL cDNA, 1 µL for both forward and reverse primers, 10 µL qPCR SuperMix (TransGen Biotech, Beijing, China), and 7 µL nucleic acid-free water. The qPCR reactive program was subjected to 39 cycles at 95 °C for 10 s, 60 °C for 10 s, 72 °C for 30 s, then 95 °C for 10 s, and 65–95 °C for 5 s. The qPCR data were analyzed by the 2^−ΔΔCt^. The means and DMRT (Duncan multiple range tests) were evaluated using SPSS software.

## Figures and Tables

**Figure 1 molecules-27-07713-f001:**
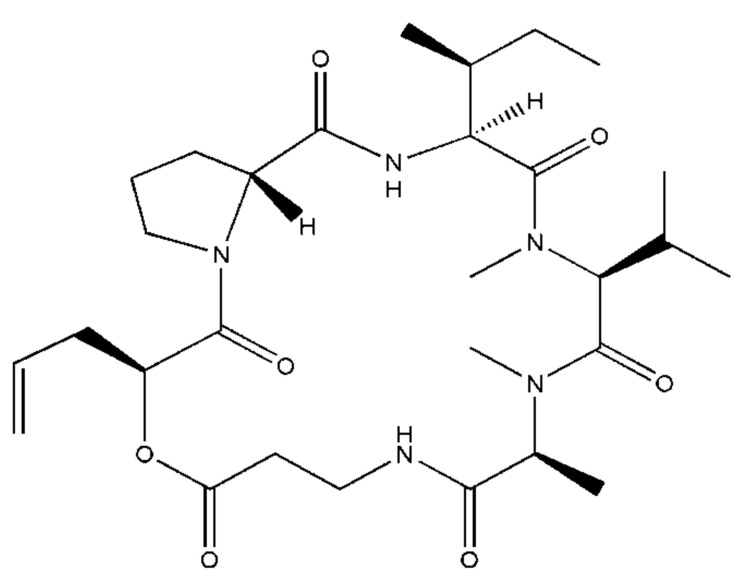
The structure of Destruxin A.

**Figure 2 molecules-27-07713-f002:**
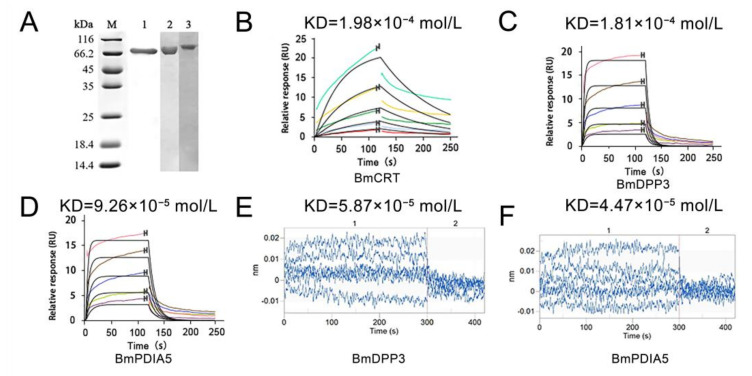
Binding affinity of DA with three proteins. (**A**) SDS-PAGE of three purified proteins. Lane M: Protein Marker; Lane 1: BmCRT; Lane 2: BmDPP3; Lane 3: BmPDIA5. (**B**–**D**) The kinetic fitting curves for the interactions between BmCRT, BmDPP3, and BmPDIA5 and DA by SPR. The curves B to D show the molecular interaction curves between proteins and different DA concentrations (250 µmol/L, 125 µmol/L, 62.5 µmol/L, 31.25 µmol/L, 15.62 µmol/L) from top to bottom. (**E**,**F**) The Align X for the interactions between BmDPP3, BmPDIA5, and DA by BLI. 1 and 2 show the processes of association and dissociation between proteins and DA.

**Figure 3 molecules-27-07713-f003:**
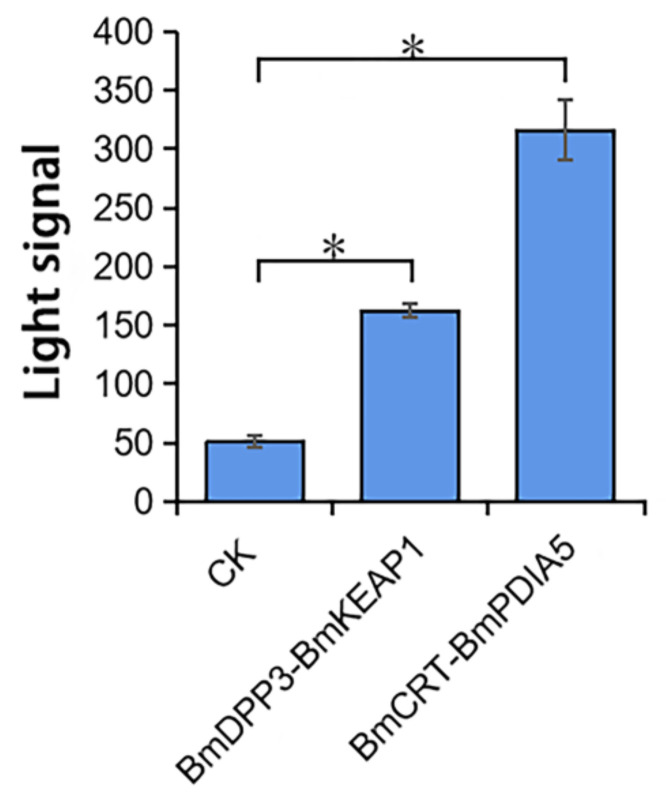
Verification of the interactions of two groups of proteins by I2H. The star (*) indicates the significant difference between two groups (*p* < 0.05).

**Figure 4 molecules-27-07713-f004:**
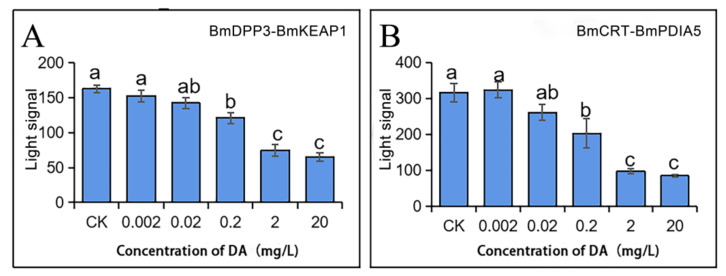
Changes in the light signal of two groups of interacting proteins after DA treatment. (**A**) BmDPP3-BmKEAP1; (**B**) BmCRT-BmPDIA5. Using the (Duncan multiple range test) DMRT to analyze, different lower case letters in the figures indicate the significant differences between groups (*p* < 0.05).

**Figure 5 molecules-27-07713-f005:**
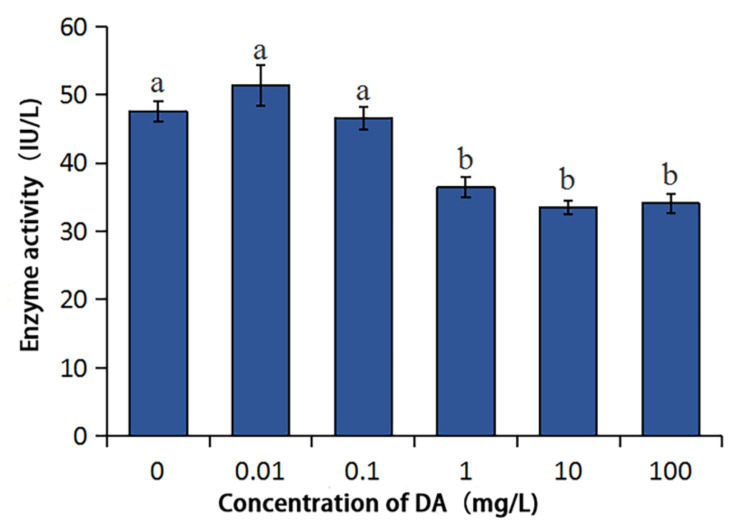
Effect of DA on the enzymatic activity of BmDPP3. Using the DMRT to analyze, different lower case letters in the figure indicate the significant differences between groups (*p* < 0.05).

**Figure 6 molecules-27-07713-f006:**
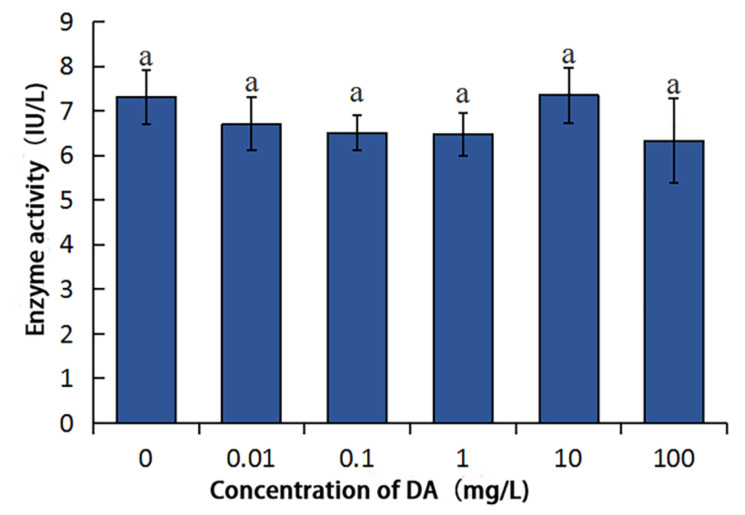
Effect of DA on the enzymatic activity of BmPDIA5. Using the DMRT to analyze, the same lower case letters in the figure indicate there were no significant differences between groups (*p* < 0.05).

**Figure 7 molecules-27-07713-f007:**
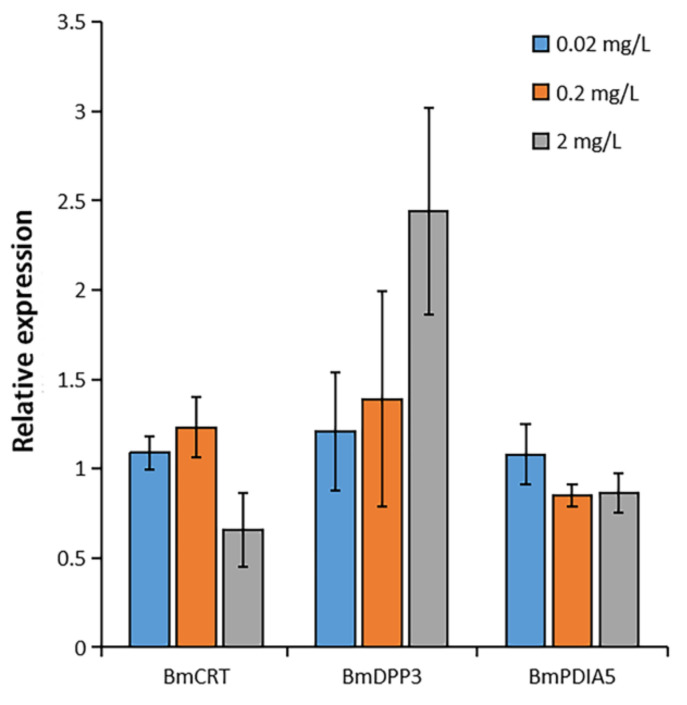
Expression of three genes after DA treatment of Bm12 cells.

## Data Availability

Not applicable.

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
