# Peer review of "Interaction of Destruxin A with Three Silkworm Proteins: BmCRT, BmDPP3, and BmPDIA5"

_molecules, 2022, doi:10.3390/molecules27227713_

Round 1
Reviewer 1 Report
I reviewed themanuscript titled “Interaction of Destruxin A with Three Silkworm Proteins, BmCRT, BmDPP3 and BmPDIA5” for your esteemed ‘MDPI Molecules”. I regret to say that this manuscript is poorly written (both language presentation and scientificquality). I am sorry to inform you that your manuscript is not acceptable in its current form
General comments
Authors investigating this study may help to understand the interaction of DA with three silkworm proteins BmCRT, BmDPP3, and BmPDIA5by 177 SPR, BLI, I2H, enzyme activity analysis, and qPCRtechniques.Which may be helpful in understanding the interaction of DA with three silkworm proteins.However, I have seen the manuscript in its current form. The manuscript is very poorly written, the authors have not mentioned the make of the chemicals and instrumentsin the materials and method section. It isconfusing (all sections) and gives the impression of deviation from the core objective of the manuscript, especially,since the discussion section is excessively irrelevant and did not properly discuss and support the findings (results) of themanuscript. The objectives of the study are not clear and are too verbose. The authors provided the figures are fuzzy and need to improve quality and resize of the figures.
My specific comments about this manuscript are given as:
A few grammatical errors in some sentences which need to be corrected. Authors are requested to go through the manuscript and correct it grammatically and Authors need to follow the spacing between the words throughout the manuscript.
In Figures : 4 You need to correct the Y-axis legends spelling in Fig: 4 A and Fig 4B
Line No: 12 “three protein” is to be corrected into “three proteins”
Line No:36 “which are” is to be corrected into “which is”
Line No:66 “proteins was” is to be corrected into “proteins were”
Line No:67 “proteins were determined by SPR is to be corrected into “proteins was determined by SPR”
Line No:106-107 “ light singal” is to be corrected into “light signal” even figure 4 A and 4B y-axis authors represented as Light singal instead of Light signal, you need to correct it in the figure 4A and 4 B
Line No:171 you need to correct the spelling of important
Line No: 212 please check the referred spelling
Line No:226 “after which the” is to be corrected into “after this enzyme activities”
Line No:241 you need to check spelling “Synergy H1 ELIASA” or “Synergy H1 ELISA”
Author Response
Dear reviewer,
I am very grateful to your comments for the manuscript. According with your advice, we amended the relevant part in manuscript. Some of your questions were answered below.
Point 1: Authors investigating this study may help to understand the interaction of DA with three silkworm proteins BmCRT, BmDPP3, and BmPDIA5 by 177 SPR, BLI, I2H, enzyme activity analysis, and qPCR techniques, which may be helpful in understanding the interaction of DA with three silkworm proteins. However, I have seen the manuscript in its current form. The manuscript is very poorly written, the authors have not mentioned the make of the chemicals and instruments in the materials and method section. It is confusing (all sections) and gives the impression of deviation from the core objective of the manuscript, especially, since the discussion section is excessively irrelevant and did not properly discuss and support the findings (results) of the manuscript. The objectives of the study are not clear and are too verbose. The authors provided the figures are fuzzy and need to improve quality and resize of the figures.
Response 1: The authors think that the suggestion of reviewer is advisable. We have added the sources of some important chemicals and instruments in the materials and methods section. And some descriptions of DA have been modified and added to the introduction section. In the abstract section, we added a sentence to indicate what the study results tried to show. We replaced the original figures with high quality figures.
Point 2: A few grammatical errors in some sentences which need to be corrected. Authors are requested to go through the manuscript and correct it grammatically and Authors need to follow the spacing between the words throughout the manuscript.
In Figures : 4 You need to correct the Y-axis legends spelling in Fig: 4 A and Fig 4B
Line No: 12 “three protein” is to be corrected into “three proteins”
Line No:36 “which are” is to be corrected into “which is”
Line No:66 “proteins was” is to be corrected into “proteins were”
Line No:67 “proteins were determined by SPR is to be corrected into “proteins was determined by SPR”
Line No:106-107 “ light singal” is to be corrected into “light signal” even figure 4 A and 4B y-axis authors represented as Light singal instead of Light signal, you need to correct it in the figure 4A and 4 B
Line No:171 you need to correct the spelling of important
Line No: 212 please check the referred spelling
Line No:226 “after which the” is to be corrected into “after this enzyme activities”
Line No:241 you need to check spelling “Synergy H1 ELIASA” or “Synergy H1 ELISA”
Response 2: The authors agree with above and revise the manuscript.

Reviewer 2 Report
1. Introduction needs to be improved to make the story about Destruxin A very clear. The first paragraph of the introduction has many information to be explained.
2. Explain, please, what is a,b,c in figure 4, 5 , 6. How are they significantly different?
3. in line 143, it is written "a zine-dependent metalloproteinase" what is a zine-dependent metalloproteinase? Do you mean a zinc-dependent metalloproteinase?
Author Response

(The authors gave the same response as above.)

Reviewer 3 Report
The manuscript entitled "Interaction of Destruxin A with Three Silkworm Proteins, BmCRT, BmDPP3 and BmPDIA5" presents original results of authors’ study, which are very interesting and could have practical value. The manuscript is suitable for publication in Molecules journal after minor revisions. Please see my comments in the attached word files.
Regards,

Author Response
Dear reviewer,
I am very grateful to your comments for the manuscript. According with your advice, we amended the relevant part in manuscript.

Round 2
Reviewer 1 Report
I review the revised manuscript“Interaction of Destruxin A with Three Silkworm Proteins, BmCRT, BmDPP3 and BmPDIA5” for your esteemed ‘MDPI Molecules” for your esteemed journal MDPI-Molecules. The study is interesting.
Authors investigating this study may help to understand the interaction of DA with three silkworm proteins BmCRT, BmDPP3, and BmPDIA5by 177 SPR, BLI, I2H, enzyme activity analysis, and qPCR techniques. Which may be helpful in understanding the interaction of DA with three silkworm proteins.The authors corrected and rewritten it in the revised manuscript as per my review comments.
Comments
We accept the revised version of the manuscript and reconsideredyour work for publication.